# Small Molecule Compounds of Natural Origin Target Cellular Receptors to Inhibit Cancer Development and Progression

**DOI:** 10.3390/ijms23052672

**Published:** 2022-02-28

**Authors:** Jinhua Wang, Dangdang Li, Bo Zhao, Juhyok Kim, Guangchao Sui, Jinming Shi

**Affiliations:** College of Life Science, Northeast Forestry University, Harbin 150040, China; jinhuawang@nefu.edu.cn (J.W.); lidd@nefu.edu.cn (D.L.); zhaobo1987@nefu.edu.cn (B.Z.); jhkim@nefu.edu.cn (J.K.)

**Keywords:** receptors, natural compounds, cancers

## Abstract

Receptors are macromolecules that transmit information regulating cell proliferation, differentiation, migration and apoptosis, play key roles in oncogenic processes and correlate with the prognoses of cancer patients. Thus, targeting receptors to constrain cancer development and progression has gained widespread interest. Small molecule compounds of natural origin have been widely used as drugs or adjuvant chemotherapeutic agents in cancer therapies due to their activities of selectively killing cancer cells, alleviating drug resistance and mitigating side effects. Meanwhile, many natural compounds, including those targeting receptors, are still under laboratory investigation for their anti-cancer activities and mechanisms. In this review, we classify the receptors by their structures and functions, illustrate the natural compounds targeting these receptors and discuss the mechanisms of their anti-cancer activities. We aim to provide primary knowledge of mechanistic regulation and clinical applications of cancer therapies through targeting deregulated receptors.

## 1. Introduction

Based on the statistics of the World Health Organization (WHO) in 2019, cancer is one of the top two causes of human death among the ages from 30 to 70 in 112 of 183 countries [1]. The GLOBOCAN 2020 estimates of cancer incidence and mortality predicted 19.3 million new cancer cases and nearly 10 million cancer deaths in 2020 [2]. In addition, a new report in 2021 also indicated that the worldwide number of cancer patients would continue to increase in the next 50 years. With the current increasing trends of major cancer types, the combinatorial incidence rate of all cancers would be doubled in 2070 compared to the number of 2020 [3]. Therefore, the global clinical and financial burdens of cancers are still very heavy. Developing novel strategies and effective therapeutics for cancer prevention and treatments are among the top priorities of all countries.

Cancer initiation and development are mediated by various signaling pathways. The first step is to receive external signals by cellular receptors, which convert exterior stimuli to intracellular signals, and trigger signaling cascades that subsequently modulate gene expression and cellular metabolism and eventually affect cell behaviors [4,5,6].

Cellular receptors can be classified into four major categories, including ion channel receptors, G protein-coupled receptors, enzyme-linked receptors and nuclear receptors, based on their structures and functions [7]. Aberrant expression and activation of receptors are general phenomena of cancer cells. Therefore, targeting receptors in cancer therapies has attracted increasing research interests, and many small chemicals, peptide agents and antibody drugs have been developed to modulate receptors’ activities.

For example, G protein-coupled receptors (GPCRs) constitute the largest membrane protein family with over 800 members. The genes encoding GPCR proteins alone occupy about 10% of the human genome [8]. The GPCR superfamily can be further divided into subfamilies or subclasses that can recognize highly diverse physiological ligands, including ions, small signaling molecules, lipids, peptides and proteins [9]. Aberrant activity of GPCRs has been associated with different diseases, and thus they are bona fide drug targets in the treatments of numerous diseases. Strikingly, up to 45% of FDA-approved drugs target GPCRs, which account for about 30% of therapeutic drugs in the global market [8,9,10]. Meanwhile, many potential GPCR agents are under clinical trials. These GPCR drugs have been used in many therapeutic areas, including cancers, hypertension, cardiovascular diseases and allergy [11].

GPCRs play a critical role in cancer initiation and progression [12]. Many studies demonstrated key roles of GPCRs in oncogenesis, including cancer initiation, progression and metastasis. GPCR mutations, such as constitutive activation of α-1B adrenergic receptor due to the substitution of three residues at its third intracellular loop [13], can confer oncogenic potential to the receptors. Meanwhile, many studies suggested that wild type GPCRs could promote oncogenic transformation due to an excess of circulating agonists or agonists produced by the tumor microenvironment. Currently, compared to GPCR-targeted therapies for many other diseases, cancer-related GPCR targets are still underrepresented, and very few GPCRs have been extensively studied for their therapeutic target potential [11]. However, the active GPCR-related studies and mounting clinical research data strongly suggest unprecedented opportunities for future applications of novel GPCR agents in cancer therapies.

The ERBB family has four protein members of receptor tyrosine kinases (RTK) that are structurally related to the epidermal growth factor receptor (EGFR). The four members in human cells include HER1 (EGFR, ERBB1), HER2 (ERBB2, neu), HER3 (ERBB3) and HER4 (ERBB4). Among them, EGFR was the first member to be identified, and also the first one linked to cancers, with the most intensive investigation [14]. Defective ERBB signaling can cause neurodegenerative diseases, such as multiple sclerosis and Alzheimer’s disease, while its excessive activation is associated with oncogenesis, especially in solid tumors [15,16]. Actually, the members of the ERBB family represent the most commonly altered proteins during oncogenesis [17]. Due to frequent overexpression, amplification or mutations, the ERBB proteins are important targets in cancer therapies [18]. Many small molecule inhibitors and monoclonal antibodies targeting the ERBB family members are currently in their clinical development and applications [18,19,20]. The molecular mechanisms of many of these drugs are also under investigation [21], and some of them are combinatorially used with other chemotherapeutics or radiotherapy to improve treatment efficacy [22].

Since ancient time, medicinal values of natural extracts, especially those from plants, have been well recognized. In recent decades, numerous natural compounds have been used as therapeutics directly and/or as templates to design and create their structural analogs for novel drug development [23]. Actually, an increasing number of natural compounds have been identified as agonists, antagonists or allosteric modulators of cellular receptors.

To date, over 600 nature-derived GPCR ligands have been discovered, and most of them are small molecules produced by plants [24]. Between 1981 and 2014, 34% of the 1562 new FDA-approved drugs were derived from natural products [24,25]. In recent years, with the rapid progress in multidisciplinary research, deep understanding of the human genome and enormous improvement of analytical technology, especially the development of high-throughput methods, interests in natural product-based drug design have been quickly resurrected. Natural products still provide the most options in drug development as both new therapeutics and effective templates for novel agents.

Natural compounds with effective anti-cancer activities are generally active substances extracted from plants and can be classified as glycosides, flavonoids, terpenoids, alkaloids and saponins [26,27]. Most of these compounds exhibit selective inhibition to cancer cells, but show mild toxicity to normal cells, and tolerable side effects to patients. They can be used alone as anti-cancer drugs [28], potentiate generic chemotherapeutics to generate synergistic effects or mitigate drug resistance [29].

Due to the barrier functions of cell membranes, chemicals with different physical structures and chemical properties are not equally permeable for cellular entry. Therefore, compounds targeting the aberrant receptors on the cell surface represent the most direct therapeutic strategy in cancer treatments. Natural compounds can modulate deregulated cellular receptors through various mechanisms. In the current review, we focused on the discussions of the natural compounds that target these receptors of the four major categories through altering their gene expression, affecting dimerization, modulating posttranslational modifications or directly binding receptor proteins. Our aim is to summarize the research and applications of different types of small molecule natural compounds targeting deregulated cellular receptors for cancer treatments and to provide a basis for improving the current therapeutic strategies.

## 2. Natural Small Molecule Compound Targeting Receptor and Its Anti-Cancer Activity

Receptors can be divided into intracellular receptors and cell surface receptors. Intracellular receptors are located in either cytoplasm or nucleus and can be triggered by membrane-permeable lipophilic hormones to exert long-term regulation in cellular activities through modulating gene expression. Cell surface receptors, also named as membrane or transmembrane receptors, are categorized as ion channel coupled receptors, GPCRs and enzyme-linked receptors, based on signal transduction mechanisms and receptor protein types. Noteworthily, upon the stimulation of ligand binding, many growth hormones, cytokine factors and their cell surface receptors can translocate into the nucleus via membrane trafficking machinery, such as endocytosis and nuclear import [30].

### 2.1. Ion Channel Coupled Receptor

Mammalian cells express a large number of ion channels that are made of pore-forming membrane proteins to selectively transport ions through the channels. These ion channels are structurally distinct and located on both the cell surface and in the membranes of intracellular organelle [31]. Ion channel receptors are modulated by different stimuli to activate their downstream signaling pathways, important for various cellular processes, including cell proliferation, differentiation and apoptosis. Therefore, altered expression or deregulated activity of ion channel receptors may generate “oncochannels” that promote cancer initiation, development and progression. Many natural compounds (Table 1) have been reported to target oncochannels, manifesting their potential as cancer therapeutics.

#### 2.1.1. Alkaloid Compounds Inhibiting Ion Channel Receptor P_2_X_7_R

P_2_X receptors (P_2_XRs) are homo- and hetero-trimeric ATP-gated cation channels with seven subtypes P_2_X_1_–P_2_X_7_ in mammals and use ATP as their physiological ligand. Many studies have demonstrated that aberrant regulation of P_2_XRs is relevant to different diseases, including cancers, neuropathic pain and arthritis [70,71]. Among these receptors, P_2_X_7_R is upregulated in various cancers, playing an important role in tumor cell invasion and metastasis, and thus considered as a potential biomarker of oncogenic transformation [72,73,74].

A number of natural compounds, such as berberine, emodin and extract from *Uncaria tomentosa*, could target P_2_X_7_R and subsequently inhibit cancer development. In recent years, inflammation has been reported as one of the major risk factors for cancer development and metastasis [75]. The P_2_X_7_R receptor, as a mediator that triggers NLRP3 inflammasome, plays a key role in metastasis related cancer cell invasion [76,77].

Berberine is a natural isoquinoline alkaloid produced by various plants, such as *Berberis vulgaris* and *Coptis chinensis* [78], and shows inhibitory activities against many types of cancers [79,80]. Lu et al. demonstrated that berberine could effectively reduce P_2_X_7_R receptor expression in oxidized low-density lipoprotein-induced macrophages [81]. Consistently, Yao et al. reported that berberine could markedly downregulate P_2_X_7_R expression in MDA-MB-231 cells, leading to reduced cell proliferation and migration [32]. Similarly, emodin, a compound isolated from rhubarb, buckthorn and Japanese knotweed, with validated anti-cancer activities [82], could also block P_2_X_7_R. A study by Jelassi B et al. demonstrated emodin as a potent P_2_X_7_R antagonist to reduce the invasiveness of human breast and lung cancer cells both in vitro and in vivo. Interestingly, compared to other P_2_XR family members, emodin selectively inhibited P_2_X_7_R-mediated signaling pathways. Consistently, the inhibitory effects of emodin were completely abolished by siRNA-mediated P_2_X_7_R knockdown [33].

The extract of *Uncaria tomentosa*, a woody vine in the tropical jungles of South and Central America [83], contains many naturally active ingredients, such as oxindole alkaloids with anti-oxidant and anti-cancer activities, and thus can induce cancer cells to undergo apoptosis [84,85]. Santos et al. reported that the hydroalcoholic extract of *Uncaria tomentosa* could reduce P_2_X_7_R receptor expression to inhibit cell proliferation and reduce progression of breast cancer, especially in combination with doxorubicin [34].

Overall, natural compounds acting as antagonists of the P_2_X_7_R receptor represent a new class of potential therapeutics to block cancer progression and improve the effects of generic chemotherapeutics.

#### 2.1.2. Flavonoids Inhibiting Ion Channel Receptor nAChRs

Nicotinic acetylcholine receptors (nAChRs) are members of the cys-loop receptor superfamily and also belong to the ligand-gated ion channel receptors. Both nicotine and its derivatives can activate nAChRs signaling pathway, leading to enhanced proliferation, migration and invasion of cancer cells [86,87]. Smoking can significantly increase the risks of several cancers, such as lung cancer, nasopharynx cancer, gastric carcinoma, bladder cancer, kidney cancer and colon cancer [88]. As a major component of cigarette smoke, nicotine is not a direct carcinogen, but it has been indicated as a cocarcinogen through receptor-dependent mechanisms, especially the activation of nAChRs, which can augment the actions of many carcinogenic ingredients inhaled by smokers. Chen et al. reported that nicotine could stimulate the nAChR and β-adrenoceptors that led to activated ERK1/2 and STAT3 signaling pathways, increased cyclin D1 expression and subsequently enhanced cell proliferation, which provided a causal connection between smoking and development of bladder and colon cancers [89,90]. In breast cancer cells, the α9-nAChR is the most abundant nAChR and has multiple downstream signaling pathways, including MAPK and PI3K/AKT [91]. With ectopic expression of α9-nAChR, nicotine could confer MCF-10A cells with malignant properties and even promote the cells to form xenograft tumors in nude mice [86].

Luteolin, quercetin and epigallocatechin-3-gallate (EGCG), which are all flavonoids, could inhibit nicotine-induced proliferation of breast cancer cells through downregulating α9-nAChR expression but showed mild adverse effect on normal mammary epithelial cells. Shih et al. also observed that siRNA-mediated knockdown of α9-nAChR could reduce the colony formation of breast cancer cells. Importantly, both luteolin and quercetin could inhibit breast cancer proliferation through blocking NFκB-mediated transcriptional activation of the α9-nAChR promoter [35]. Interestingly, the combination of siRNA-mediated α9-nAChR knockdown and the natural compounds showed more dramatic inhibition to breast cancer cells than their individual treatments, strongly suggesting that luteolin and quercetin may have additional targets besides α9-nAChRs [35]. Similarly, EGCG also showed antagonistic effects on α9-nAChR activation. In a report by Tu et al., the α9-nAChR promoter activity was significantly induced by both nicotine and estradiol, but the induction was virtually abolished by the pretreatment of EGCG [36]. In addition to flavonoids, garcinol, a polyisoprenylated benzophenone derivative, could also antagonize α9-nAChR to inhibit breast cancer cell proliferation through a mechanism of antagonizing α9-nAChR-mediated cyclin D3 expression [37].

Overall, these data indicated that α9-nAChR and its related receptors are promising targets for cancer therapies. Natural compounds luteolin, quercetin, EGCG and garcinol are effective candidates of anti-cancer therapeutics through modulating these deregulated receptors.

### 2.2. Enzyme-Linked Receptor-Tyrosine Kinases

#### 2.2.1. Small Natural Compounds Inhibiting Epidermal Growth Factor Receptor

Aberrant expression and activation of the ERBB proteins have been frequently reported during malignant transformation and associated with poor prognoses. Among these receptors, EGFR and HER2 are both well recognized biomarkers and genuine therapeutic targets of different cancers, but overexpression of HER3 and HER4 may lead to differential prognostic outcomes of cancer patients [92,93,94]. The ERBB receptors can activate a variety of signaling pathways regulating cell proliferation, invasion, angiogenesis, apoptosis and differentiation [95,96,97]. EGFR mutations are frequently observed in lung cancers and related to shorter progression-free survival [98]. The HER2/neu gene is amplified and overexpressed in 25–30% of human breast and ovarian cancers and associated with poor prognoses of the patients [99].

Many natural compounds have been reported to modulate ERBB signaling pathways, and thus manifest therapeutic potential (Table 1).

Silibinin, taspine and resveratrol reportedly bind to EGFR to block its activation and subsequently dampen its downstream signaling pathways in cancer cells. Silibinin is present in the extract of the *milk thistle* seeds [100]. It can directly bind EGFR, decrease its phosphorylation and subsequently inhibit the expression of its downstream target lysyl oxidase (LOX), leading to reduced migration of non-small cell lung cancer (NSCLC) cells. Mechanistically, silibinin could form a π-π stacking interaction with the F723 residue and hydrophobic interactions with other four residues of EGFR to strongly bind the receptor and block its activity. Interestingly, the binding site of silibinin to EGFR is different from that of oxitinib, one of the third-generation epidermal growth factor receptor tyrosine kinase inhibitors (EGFR-TKIs), suggesting that silibinin may also be effective on cancer cells harboring the EGFR mutant resistant to the EGFR-TKI [38].

Another natural compound showing similar inhibition against EGFR is taspine that is produced in various plants, such as *Magnolia* × *soulangeana* and *Croton lechleri* [101,102]. Taspine could antagonize the EGFR signaling pathway to reduce cell proliferation and migration and inhibit xenograft tumor formation. A molecular docking study suggested that taspine formed six hydrogen bonds with five residues in the binding pocket of the EGFR protein [39]. As a star medicinal molecule, resveratrol, mainly extracted from grapes, is a favorite natural compound for both nutritional purposes and anti-cancer therapeutic treatments [103].

Resveratrol, as a polyphenol, has been shown to exert inhibitory activity against many cancer types, especially through the mechanism of targeting deregulated EGFR signaling pathway in cancer cells [104]. In vivo, resveratrol markedly suppressed mouse xenograft tumor growth of NSCLC cells [105]. Moreover, resveratrol could be used as a sensitizing agent. As an example, resveratrol synergized gefitinib, one of the first-generation EGFR-TKIs, to inhibit the proliferation of gefitinib-resistant NSCLC cells [106]. Mechanistically, in the cotreatment of the two compounds, resveratrol could increase the intracellular accumulation of gefitinib and subsequently inhibit EGFR phosphorylation in NSCLC cells. With concomitantly increased expression of cleaved caspase 3, p53 and p21, the cells would undergo apoptosis, autophagy, cell cycle arrest and senescence [106].

In prostate cancer therapies, one of the major challenges in the treatment of advanced prostate cancer is the lack of the effective remedy for hormone refractory tumors, which are mostly resistant to the androgen deprivation therapy [107]. A study by Wang et al. revealed that resveratrol could directly bind EGFR to rapidly dampen its phosphorylation, leading to reduced AKT activation in prostate cancer cells. These results indicated that resveratrol could inhibit the proliferation of both AR positive and negative prostate cancer cells through AR-dependent and -independent pathways [40]. With many additionally reported anti-cancer activities borne in mind, it is reasonable to predict that resveratrol is a superior candidate for the treatments of hormone refractory prostate cancer.

Natural compounds are frequently used together with other common anti-cancer agents to achieve synergistic inhibition and/or reduce drug resistance. Honokiol, isolated from the bark, seed cones and leaves of the trees of the genus *Magnolia*, could inhibit EGFR in head and neck squamous cell carcinoma (HNSCC) cells, resulting in reduced activities of its downstream proliferative and survival targets, such as AKT, STAT3, BCL-XL and cyclin D1. Importantly, honokiol could significantly improve the anti-proliferative and anti-invasion activities of erlotinib, a broadly used EGFR inhibitor, and eliminate the resistance to erlotinib of the HNSCC cells. In addition, the combinatory use of honokiol and cetuximab, a chimeric antibody against EGFR, could effectively inhibit the xenograft tumor formation [41].

Genistein is an isoflavone discovered in soy and possesses many medicinal activities, including inhibiting inflammation, promoting apoptosis and modulating steroidal hormone receptors [108]. Importantly, genistein was also reported to block RTKs, including EGFR. A study by Park et al. demonstrated that the combination of genistein and cetuximab could significantly inhibit the growth of xenograft tumors generated by oral squamous cell carcinoma (OSCC) cells, but individual use of these two agents did not show inhibitory effect on tumor development. However, in vitro studies using two OSCC cell lines indicated that genistein failed to potentiate the anti-cancer activity of cetuximab or only show additive effect [42].

Ligand-dependent dimerization of EGFR, as well as many other RTKs, is an essential step of activation for the receptor. However, mutations of the EGFR protein could promote its dimerization in a ligand-independent manner [109]. In lung cancer cells, silibinin could selectively reduce the dimerization of EGFR mutants and subsequently dampen their activities, but wild type EGFR was not affected. Therefore, the combination of silibinin and erlotinib could effectively inhibit the growth of xenograft tumors formed by erlotinib-resistant lung cancer cells [43].

In HNSCC, EGFR overexpression is frequently observed and associated with poor prognoses of the patients. EGCG was shown to inhibit EGFR phosphorylation and reduce the activation of its downstream effectors, including STAT3, ERK proteins c-fos and cyclin D1. Furthermore, EGCG could greatly potentiate the growth inhibition of HNSCC caused by 5-fluorouracil, a generic cancer therapeutic [44].

These results strongly suggest that honokiol, genistein, silibinin and EGCG can be used as effective therapeutic agents in cancer treatments to augment the effects of generic EGFR inhibitors and alleviate patients’ resistance to these conventional drugs.

Both EGFR and HER2 increases are also common phenomena in ovarian cancer and are associated with adverse clinical outcomes. In ovarian cancer cells, berberine could reduce the expression of both EGFR and HER2 and suppress their targets, cyclin D1, matrix metalloproteinases and vascular endothelial growth factor (VEGF). Interestingly, berberine could apparently promote HER2 ubiquitination and proteasome degradation, uncovering a novel mechanism of berberine-mediated receptor downregulation [45]. EGFR is also an oncogenic marker of colorectal cancer. Wang et al. demonstrated that berberine could stimulate the E3 ligase activity of the CBL protein to enhance EGFR ubiquitination and degradation, leading to G1/S and G2/M cell cycle arrest of colon cancer cells. Consistently, depletion of CBL by siRNA abolished berberine-mediated inhibition of EGFR activity. The results indicate that berberine can be used as a dual-blocker of both EGFR and HER2 to ensure effective treatment of highly metastatic cancers with EGFR and/or HER2-positive statuses and reduce their drug resistance [110].

#### 2.2.2. Small Natural Compounds Inhibiting Vascular Endothelial Growth Factor Receptors

The signaling pathways stimulated by vascular endothelial growth factors (VEGFs) mainly regulate angiogenesis, lymphangiogenesis and vascular permeability. The VEGF-mediated signaling is also present in cancer cells and contributes to cancer stem cells and tumor initiation [111]. The classical VEGF receptors (VEGFRs) include VEGFR1, 2 and 3. Among them, VEGFR2 is the predominant RTK that mediates VEGF signaling to promote angiogenesis and therefore is considered as a major target in anti-angiogenic treatments in different cancer therapies [112]. Many small molecule inhibitors of VEGFR2 are currently in clinical trials, but the long-term usage of these drugs could generally lead to intolerable side effects [113]. Therefore, it is necessary to develop inhibitors of natural origin as anti-angiogenic therapeutics (Table 1).

Voacangine is an alkaloid compound isolated predominantly from the root barks of many plants. Kim et al. demonstrated that voacangine could bind the kinase domain of VEGFR2, using the drug affinity responsive target stability (DARTS) method, which was based on improved stability of a protein when interacting with a natural compound. Consistently, voacangine could block the kinase activity of VEGFR2 through docking simulation and inhibit cell proliferation of glioblastoma cells expressing high levels of VEGFR2. Importantly, using the mass spectrometry imaging (MSI) method, the authors detected intensified overlap between voacangine and xenografted glioblastoma tumor compartments in nude mice [46].

Ellagic acid is a polyphenol present in many fruits and vegetables. Wang et al. used molecular docking simulation to demonstrate that ellagic acid could form hydrogen bonds and aromatic interactions with the ATP-binding region of VEGFR2. Consistently, ellagic acid could inhibit VEGF-induced angiogenesis, block VEGFR2 kinase activity and dampen its downstream signaling pathways, including MAPK and PI3K/AKT in cancerous endothelial cells [47]. In various bladder cancer cell lines, ellagic acid could inhibit VEGFR2 expression, likely through promoting its degradation or reducing its synthesis. In line with this regulation, ellagic acid inhibited bladder cancer cell proliferation and invasion [48].

The combinatorial uses of natural compounds with common cancer therapeutics have drawn increasing attention and may become general practices in cancer therapies. The uses of natural compounds could sensitize cancer cells to conventional chemotherapeutics, and thus decrease their administration dosages and frequencies, leading to reduced or tolerable side effects.

The combination of voacangine with sunitinib could significantly inhibit VEGFR2 activity likely through reducing glioblastoma cells’ resistance to sunitinib [46]. Synergistic inhibitory effects could be achieved when ellagic acid and mitomycin C, a generic anti-cancer drug, were used together in the treatment of bladder cancer cells [48]. Additionally, the combination of matrine, a compound found in plants of the genus *Sophora*, and cisplatin could inhibit a number of proliferative and survival signaling pathways in urothelial bladder cancer cells, including downregulation of VEGFR2 expression. The combined administration of matrine and cisplatin can enhance the sensitivity of urothelial bladder cancer cells to cisplatin, reduce its dosage and mitigate its side effects, indicating that matrine can potentially be used as an adjuvant drug in the therapies of urothelial bladder cancer [49].

#### 2.2.3. Flavonoids and Polyphenols Inhibiting FLT3, c-Met, EphA2R

Fms-like tyrosine kinase 3 (FLT3), also known as the cluster of differentiation antigen 135 (CD135), is a cytokine receptor belonging to the RTK family and expressed on the surface of hematopoietic progenitor cells [114,115]. Both overexpression and overactive mutations of FLT3 have been implicated in different diseases, prominently in acute myeloid leukemia (AML), of which one-third of patients carry FLT3 mutants that correlated with poor prognosis [114]. Ponatinib, midostaurin and quizatinib, as kinase inhibitors of FLT3, have been used in the treatment of AML cells harboring mutated FLT3 [116,117,118]. However, clinical data showed that the F691L mutation of FLT3 would confer cancer cells with acquired resistance to certain FLT3 inhibitors [119].

Isoliquiritigenin (ISL), a flavonoid compound found in licorice, could effectively inhibit the kinase activity of FLT3 with an IC_50_ value at a nanomole/liter level (Table 1). ISL selectively blocked the proliferation of the AML cells with mutated FLT3, but not the cells lacking these mutations, and subsequently reduced the activities of FLT3-mediated downstream pathways. In a mouse model, oral administration of ISL suppressed the growth of the tumors xenografted by the AML cells. Consistently, molecular docking analysis also showed that ISL could bind the kinase structural domain of FLT3 through aromatic interactions and hydrogen bonds. These results suggest that ISL is a promising bioactive compound that can be used in the treatment of AML patients [50].

Erythropoietin-producing hepatocellular carcinoma (Eph) receptors represent a major subfamily of the RTKs and are involved in angiogenesis and tumor progression. Among these receptors, the EphA2 receptor is commonly overexpressed in tumor tissues, but its levels in normal tissues are generally low, suggesting its potential as a target in cancer therapies [120]. Giorgio et al. tested 21 phenolic compounds and found that urolithin D blocked ephrin A1-mediated EphA2 phosphorylation, but its biological effects remained unclear [51] (Table 1).

c-Met also belongs to the RTK family, and HGF and its alternative splicing isoforms represent the only known endogenous ligands of c-Met. The aberrant activation of c-Met promotes tumor growth, angiogenesis and metastasis, and thus correlates with poor prognosis of cancer patients [121]. Quercetin could inhibit HGF-stimulated melanoma cell migration and invasion through reducing c-Met phosphorylation, dimerization and gene expression, leading to the inhibition of a number of c-Met downstream targets [52] (Table 1). Moreover, many other known flavonoids, such as EGCC and luteolin, also showed inhibitory effects on HGF-stimulated cancer cell migration [122,123]. Therefore, these plant-derived flavonoids possess common activities to target deregulated receptors and can be potentially used in both cancer prevention and treatment.

### 2.3. G Protein-Coupled Receptors

#### 2.3.1. Small Natural Compounds Inhibiting Protease-Activated Receptor-2

Protease-activated receptors (PARs) belong to the seven transmembrane domain GPCRs, and include four members, PAR1-4, that play important roles in normal physiological processes and various diseases, including cancers [124]. The four PARs can be differentially activated by thrombin (for PAR1, PAR3 and PAR4) or trypsin-like serine proteases (for PAR2) [125]. Among these receptors, PAR2 activation contributes to cancer cell motility and metastasis [125,126], suggesting its therapeutic target potential for cancer patients.

A study by Kakarala et al. screened multiple phytochemicals for their inhibitory activity against PAR2 in breast cancer. Among these tested phytochemicals, EGCG exhibited the strongest binding affinity to PAR2 [127]. Consistently, in human colon cancer cells, EGCG could block the proliferation and migration induced by a peptide agonist of PAR2 [53] (Table 1). These data suggested that EGCG could inhibit cancer development and progression through targeting the PAR2 signaling pathways.

#### 2.3.2. Small Natural Compounds Inhibiting Prostaglandin E Receptors

Prostaglandin (PG) receptors, or prostanoid receptors, are cell surface membrane GPCR receptors for different prostanoids. Among them, E-type prostanoid (EP) receptors have four members, including EP1-4 receptors [128]. Masato et al. reported that EP1, EP2 and EP4 played important roles in prostate oncogenic transformation in patients with hormone sensitive prostate cancer [129]. Among the activating ligands, prostaglandin E2 (PGE2) is an important cell growth and regulatory factor to stimulate the EP2 receptor and exert oncogenic activities, including enhancing angiogenesis, promoting tumor invasion and metastasis and causing multidrug resistance [130].

Silymarin, EGCG and frondoside reduced PGE2-induced EPR expression and basal EPR levels. They inhibited PGE2-induced cancer migration or metastasis through inhibition of EPR signaling pathways (Table 1).

Silymarin is a standardized extract of the milk thistle seeds and exhibits various anti-cancer activities, including inducing apoptosis, inhibiting cell proliferation and blocking cancer cell migration and invasion. Woo et al. reported that silymarin inhibited EP2 receptor-mediated activation of its downstream targets, including the phosphorylation of CREB, Src and STAT3, leading to retarded migration of renal cell carcinoma cells. The specific inhibition of silymarin was proved by its antagonistic effects against the treatment of Butaprost, an agonist of the EP2 receptor [54]. In a study by Jin et al., the authors observed increased EP1 receptor expression in hepatocellular carcinoma cells compared to the normal hepatocytes. Additionally, EGCG could target the EP1 receptor and subsequently inhibit the viability and migration of the cancer cells [55].

Frondoside A is a triterpenoid saponin isolated from the sea cucumber Cucumaria frondose and has been demonstrated to have potent cytotoxicity in different cancers, such as pancreatic cancer and triple-negative cancer cells. Ma et al. revealed a mechanism of frondoside A to antagonize the activities of the EP2 and EP4 receptors in breast cancer cells. Consequently, frondoside A blocked the EP4/EP2-mediated intracellular cAMP activation and EP4-mediated ERK1/2 activation, leading to reduced spontaneous metastasis of breast cancer cells to the lungs in a mouse model [56].

In cancer cells, cyclooxygenase 2 (COX-2) is a key regulator driving the synthesis of PGE2, which subsequently promotes oncogenesis through binding to its receptors; thus, targeting COX-2 has been generally employed in cancer therapies. However, extended uses of COX-2 inhibitors have been associated with significant cardiotoxicity of cancer patients [131]. Therefore, inhibition of PG-activated EP receptor signaling is considered as an alternative and practical approach for halting tumor progression [132]. Based on these clinical circumstances, silymarin and frondoside A targeting PG-activated EP receptor signaling can be used as substitutes of COX-2 inhibitors in cancer therapies to reduce side effects and improve therapeutic efficacies.

#### 2.3.3. Flavonoids Inhibiting Cannabinoid Receptors

Cannabinoid classical receptors, including CB1 and CB2, are 7-transmembrane receptors that mediate the central and peripheral actions of extracts from *Cannabis sativa*. CB1 receptor is highly expressed in the human central nervous system, and its ligands, both agonists and antagonists, have been used to treat various diseases [133,134]. In colorectal cancer cells, quercetin treatment could significantly increase the expression of the CB1 receptor, leading to inhibited cell proliferation and migration. Concurrently, quercetin could inhibit the PI3K/AKT/mTOR signaling pathway and induce the proapoptotic JNK/JUN pathway [58].

Delta-9-tetrahydrocannabinol (THC) and cannabidiol (CBD) are two cannabinoids approved by the Food and Drug Administration (FDA) of the United States [135]. The activation of the cannabinoid receptors CB1 and CB2 by THC can inhibit cell proliferation and invasion, induce apoptosis of cancer cells and reduce tumor growth in vivo [57,136,137]. In addition, THC has been used for the alleviation of adverse symptoms caused by chemotherapies, such as vomiting, nausea and pain; however, high doses of its usage can cause psychological side effects, such as compromised performance of memory, attention and reaction time, although these symptoms will mostly vanish over time [135,138]. Nevertheless, the cannabinoids are still considered as relatively safe in their clinical applications, because these adverse effects are within the acceptable range compared to other drugs, especially in cancer therapies, and may fade away with continuous uses.

Importantly, the psychological side effects of cannabinoids are mostly mediated by CB1, but not CB2, receptors in the brain. Therefore, the cannabinoids that selectively target CB2 receptors should be preferentially considered in cancer therapies [139]. Actually, CBD is a non-psychoactive cannabinoid, and thus can be further developed for its application in cancer treatment [140]. Although CBD showed relatively low binding affinity to both CB1 and CB2 receptors, it could activate the two receptors independently to exert anticancer effects [141,142]. Therefore, the combinatorial use of THC and CBD manifested better anticancer activities than THC alone, which would also reduce the dosage and side effects of THC [57,142,143]. Marcu et al. reported that THC synergistically cooperated with CBD to inhibit the proliferation of glioblastoma cells, with reduced dosage of THC [57]. Interestingly, CBD has also been indicated to alleviate some adverse effects caused by THC, such as convulsions, discoordination and psychiatric symptoms, which may improve the tolerability of cannabis-based medicines [142,143]. In addition, the combination of low-dose THC and CBD with temozolomide, a generic drug against glioblastoma, could significantly reduce xenograft tumor growth of glioma cells [144].

#### 2.3.4. Small Natural Compounds Inhibiting Frizzled Receptors

The Frizzled (FZD) is a family of atypical GPCRs and consists of 10 members (FZD1-10). The general Wnt/β-catenin signaling cascade can be mediated through the Wnt ligand-mediated FZD receptors and lipoprotein receptor-related protein 5/6 (LRP5/6) receptors. The aberrant expression and regulation of these receptors are associated with oncogenesis. Among them, the FZD7 receptor is frequently overexpressed in various cancers, and considered as a therapeutic target [145,146], while the FZD8 receptor possesses similar activities, and contributes to drug resistance of cancer cells [147].

The ATP-binding cassette (ABC) transporters confer cancer cells resistance to cytotoxic and targeted chemotherapies [148]. Chen et al. reported that quercetin treatment could reduce FZD7 levels and subsequently dampen the Wnt/β-catenin signaling pathway. Through this mechanism, quercetin could both downregulate the expression and inhibit the function of the ABC transporters, ABCB1, ABCC1 and ABCC2, leading to reversal effects on multidrug resistance [59]. These data suggest that quercetin attenuates chemoresistance of cancer cells at least partially through inhibiting the FZD7 receptor, and thus can potentially be used as a natural sensitizer to treat drug-resistant cancer cells.

Brucine is an alkaloid commonly present in the *Strychnos nux-vomica* tree and has been reported to possess anti-cancer activities in a variety of cancers [149]. A report by Shi et al. indicated that brucine could downregulate the expression of FZD8 and decrease the phosphorylation of LRP5/6 receptor to induce apoptosis, inhibit growth and migration and block xenograft tumor formation of colorectal cancer cells. Therefore, brucine is a promising candidate therapeutic to suppress colon cancer progression [60].

### 2.4. Nuclear Receptor Superfamily

#### Polyphenols Inhibiting Estrogen Receptor and Androgen Receptor

Nuclear receptors, located in cells, can sense specific molecules, such as steroid and thyroid hormones, and consequently regulate the expression of specific genes involved in different biological processes, such as development, homeostasis and metabolism. Most nuclear receptors directly bind DNA and act as transcription factors to regulate gene expression, which makes them different from other receptors.

The estrogen receptors (ERα and ERβ) and the androgen receptor (AR) are members of the nuclear receptor superfamily. When estrogen and androgen attach to the ER and AR, respectively, the receptors can directly bind to the DNA at their corresponding responsive elements, or associate with other transcription factors, such as AP1 and NFκB, to modulate numerous biological processes of normal development, differentiation, proliferation and oncogenesis [150,151]

Many plant-derived polyphenols, including flavonoids and phytoestrogens, can act as either activators or antagonists to regulate ER and AR activities and their downstream pathways (Table 1). Ellagic acid possesses several phenolic rings and ortho-dihydroxy groups, structurally resembling estrogen. Therefore, ellagic acid can be recognized by ellagic acid to mediate its action, and thus act as a natural selective estrogen receptor modulator. At the concentrations of nanomole per liter levels, ellagic acid displayed a significant estrogenic activity through activating ERα but acted as an estrogen antagonist to ERβ [61]. Chiu et al. reported that luteolin promoted proteasome-mediated AR degradation through disrupting its association with heat-shock protein 90 (HSP90). Consistently, luteolin inhibited prostate cancer through an androgen-dependent manner; it could significantly reduce proliferation and induced apoptosis of LNCaP cells, an androgen-dependent cell line, but showed much less effects on two androgen-independent cell lines PC-3 and DU145. Additionally, luteolin could inhibit xenograft tumor formation by LNCaP cells [62]. Ablation therapy is a common strategy to treat advanced prostate cancer, but many patients may eventually relapse with castration-resistant prostate cancer [152]. In these relapsed tumors, increased AR levels are frequently detected [153], and thus preventive approaches against AR expression or activity can reduce the morbidity and mortality of the frustrated prostate cancer. The flavonoid compounds discussed above can be used as natural medicinal agents to dampen AR levels or activities, and thus play both chemopreventive and chemotherapeutic roles against prostate cancer [154].

Estrogen-related receptor gamma (ESRRG) is another member of the nuclear receptor superfamily. Currently, ESRRG is an orphan receptor without any validated physiological activating ligand. In many cancers, ESRRG exhibits tumor suppressor activities [155,156]. Sophoridine (Table 1), a bioactive alkaloid present in many Chinese herbs [157], can potentiate the tumor suppressive activity of ESRRG in gastric cancer cells. Peng et al. reported that sophoridine could promote β-catenin degradation through enhancing ESRRG expression and subsequently inhibit gastric cancer cell proliferation, colony formation, migration and invasion and induce apoptosis. Furthermore, sophoridine could cause cell cycle arrest at the G2/M phase through inhibiting DNA repair and enhance the efficacy of cisplatin, suggesting its promising therapeutic potential in gastric cancer treatments. Therefore, sophoridine can potentially become an effective ESRRG inducer to suppress cancer progression [63].

### 2.5. Other Receptors

#### 2.5.1. Small Natural Compounds Combined with TRAIL to Inhibit the Death Receptor DR5

Tumor necrosis factor (TNF)-related apoptosis-inducing ligand (TRAIL) is a member of the TNF superfamily that can initiate the apoptosis pathway through binding to its death receptors DR4 and DR5 [158,159]. Both curcumin and compound K (CK) can induce the expression of DR5 to activate the TRAIL pathway (Table 1).

Curcumin is a natural diketone found in *Curcuma longa* species and possesses anti-inflammatory and anti-cancer activities [160]. In human renal cancer cells, curcumin could markedly induce DR5 expression through a mechanism of increasing the levels of reactive oxygen species (ROS), leading to apoptotic cell death. Thus, in terms of a clinical perspective, the combination of curcumin and TRAIL represents a novel strategy for the treatment of a variety of human cancers that are resistant to conventional chemotherapies. However, oral administration of curcumin showed poor bioavailability due to its rapid metabolism in the liver and intestine. Therefore, further investigation is needed to improve in vivo stability and bioactivity of curcumin [64].

Ginsenoside compound K is an active metabolite of ginseng or ginsenoside. Chen et al. demonstrated that ginsenoside compound K upregulated DR5 expression through both autophagy-dependent and -independent mechanisms, leading to ameliorated TRAIL resistance and improved TRAIL-induced apoptosis in colorectal cancer cells. The results suggested synergistic anti-cancer activity with the combinatorial treatment of ginsenoside compound K and TRAIL in colorectal cancer [65].

#### 2.5.2. Small Natural Compounds Inhibiting Peroxisome Proliferator-Activated Receptors and Toll-Like Receptor 4

Peroxisome proliferator-activated receptors (PPARs) constitute a family of nuclear receptors including three subtypes, PPARα, PPARγ and PPARβ/δ [161]. Among them, PPARγ is a major transcription factor in adipocytes and primarily regulates lipid metabolism and bioenergetic requirements in living systems. Aberrant PPARγ expression often accompanies cancers [162]. Ginsenoside F2 (Table 1) could directly bind PPARγ to reduce its expression, leading to reduced adipocyte differentiation. Through molecular docking analysis, ginsenoside F2 could form hydrogen bonds with the Y473 residue of the PPARγ protein [66].

Thymoquinone is a phytochemical compound present in the extract of the plant *Nigella sativa* [163], and its anti-cancer activity has been reported by different research groups, including ours [164]. Woo et al. demonstrated that thymoquinone could act as a natural ligand to directly bind and activate PPARγ. Mechanistically, thymoquinone could interact with seven polar residues and six non-polar residues within the ligand-binding pocket of PPARγ to activate the receptor. Consequently, thymoquinone treatment could lead to downregulation of BCL-2, BCL-XL and survivin and induce apoptosis of breast cancer cells [67]. This study built the connection between the anticancer activities of thymoquinone and the activation of the PPARγ pathway through direct compound–receptor interaction, which led to altered expression of downstream target genes. Further investigations are necessary to develop effective and practical strategies of thymoquinone-based cancer prevention and therapies.

Another natural compound showing inhibitory activity to PPARγ is 6-shogaol, a dehydrated product of 6-gingerols that can be extracted from fresh ginger [165]. A study by Isa et al. revealed that both 6-shogaol and 6-gingerol could inhibit TNF-α-mediated downregulation of adiponectin expression in mouse adipocytes through activation of PPARγ [166]. Tan et al. reported that 6-Shogaol could both activate the transcription of the PPARγ gene and directly bind to the PPARγ receptor with binding affinity comparable to that of a PPARγ natural ligand. Consequently, 6-shogaol could induce apoptosis of breast and colon cancer cells [68].

Toll-like receptors (TLRs) are single-pass membrane-spanning receptors that include 13 members, TLR1-13, and play key roles in the innate immune system [167]. Aberrant regulation of TLRs has been frequently reported to be involved in oncogenesis, such as the participation of TLR3 in neuroblastoma, breast cancer, hepatocellular carcinoma, nasopharyngeal cancer and lung cancer [168], and the overexpression of TLR4 in lung cancer, neuroblastoma, colorectal cancer and thyroid cancer [169]. Dimerization of the receptor is a prerequisite for the activation of many receptors to trigger their downstream pathways. Youn et al. reported that curcumin could block TLR4 dimerization in both ligand-induced and ligand-independent manners and inhibit the activation of its subsequent signaling pathways [69]. Another report by Bppzari et al. also indicated the antagonistic effects of curcumin on TLR2, 4 and 9, strongly suggesting potent therapeutic activities of curcumin in cancer therapies [170]. However, Li et al. reported that curcumin-induced intracellular ROS generation could stimulate TLR4 expression, leading to apoptosis of liver cancer cells [171]. Therefore, the clinical applications of curcumin in the treatments of cancers and other diseases through regulating TLRs deserve further investigations.

## 3. Conclusions

Small molecule compounds of natural origin have been widely used as chemotherapeutic and adjuvant agents in cancer treatments due to their effective anti-cancer activities and relatively low side effects. This review summarizes the use of many natural compounds as inhibitors or activators to target different receptors of cancer cells (Figure 1).

The functionality of small natural molecules in targeting the receptors to regulate oncogenic processes depends on their structure and properties. For example, alkaloids, flavonoids and cyclic monoterpenoids, as well as their derivatives, can effectively inhibit ion channel protein receptors [172]. Plant-origin polyphenols, including flavonoids and phytoestrogens, are structurally similar to estrogens and can thus modulate nuclear hormone receptors [173]. Derivatives of berberine have geometric structures related to aromatic interaction-based molecular recognition in biological systems and can subsequently regulate corresponding receptors [174]. Ellagic acid bears phenolic rings and ortho-dihydroxy groups, which resembles estrogen and thus modulates the action of ER [61]. Angiogenesis is indispensable to excessive tumor growth; therefore, natural compounds capable of antagonizing VEGFRs, especially VEGFR2, can inhibit oncogenic processes. Phenolic compounds can inhibit cancer through epigenetically upregulating DR5, leading to the activation of the TRAIL apoptotic pathway [175].

Some natural compounds can target different types of cellular receptors, and their applications may effectively inhibit cancer development and progression through simultaneously deactivating multiple signaling pathways. For example, genistein blocks EGFR and ER to suppress cholangiocarcinoma [176]. Targeting two functionally different receptors and their downstream signaling pathways represents effective and promising strategies in developing novel cancer therapeutics [177,178].

Natural compounds inhibit cellular receptors through different mechanisms, including downregulating receptor gene expression, directly binding receptors to antagonize their activation, interfering with receptor dimerization, altering receptor modifications, especially phosphorylation and promoting receptor degradation. In some cases, these regulatory mechanisms may show crosstalk or interplay. In functional studies, some natural compounds may potentiate or synergistically improve generic therapeutics to inhibit cancer development and progression.

In this review, we have summarized different types of small molecule compounds of natural origin targeting cellular receptors for cancer therapies to provide a basis for the development of natural anti-cancer drugs and assist the improvement of current therapeutic strategies.

## Figures and Tables

**Figure 1 ijms-23-02672-f001:**
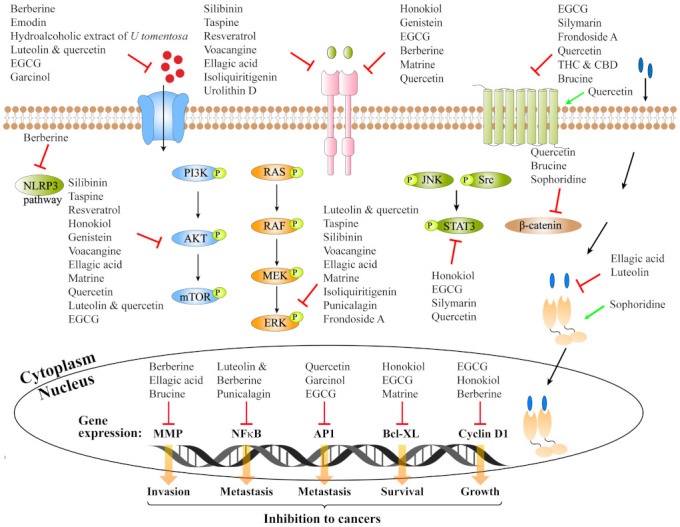
Small molecule compounds of natural origin and their targeted cellular receptors. The regulation of the proteins and signaling pathways on cell membrane, in cytoplasm and the nucleus by the small molecule compounds has been described in detail in the main text.

**Table 1 ijms-23-02672-t001:** A summary of small molecule compounds of natural origin targeting different receptors and signaling pathways involved in cancer development and progression.

Receptors	Targets	Compounds	Compound Categories	Main Sources	Biological Models	Pharmacological Activities	Solvents	Dosages	Ref.
Ion-channel-coupled	P_2_X_7_R	Berberine	Isoquinoline -alkaloid	*Huanglian* and *Huangbai*	MDA-MB-231	Reduce P_2_X_7_R and NLRP3 expression; anti-metastasis	Water-soluble	2.5–100 μg/mL	[32]
		Emodin	Anthraquinone	*Rheum- palmatum*	MDA-MB-435	Inhibit P_2_X_7_R-dependent Ca^2+^ signaling and ECM degradation; anti-metastasis	DMSO	10 μM	[33]
		Hydroalcoholic extract *U tomentosa*	Oxindole alkaloids	*Uncaria tomentosa*	MDA-MB-231	Reduce P_2_X_7_R expression; anti-metastasis	60% ethanol	250 and 500 μg/mL	[34]
	α9-nAChR	Luteolin and Quercetin	Flavonoid	*Sophora and Japonicapeanut*	MDA-MB-231	Reduce α9-nAChR expression; anti-proliferation	DMSO	0.5–1 μM	[35]
		Epigallocatechin-3-gallate	Flavonoid	*Camelia sinensis*	MCF-7	Reduce α9-nAChR expression; anti-proliferation	DMSO	1–10 μM	[36]
		Garcinol (Gar)	Polyisoprenylated benzophenone	*Garcinia indica fruit rind*	MDA-MB-231 and MCF-7	Reduce α9-nAChR and cyclin D3 expression; anti-proliferation	DMSO	1–10 μM	[37]
Enzyme-linked receptor	EGFR	Silibinin	Polyphenolic	*Silybium marianum*	Head and neck squamous cell carcinoma	Bind to EGFR, inhibit EGFR phosphorylation; inhibit EGFR/LOX pathway; anti-migration; anti-metastasis	-	5–20 μM	[38]
		Taspine	Alkaloid	*Magnolia spec.*	A431 epidermoid cancer and HEK293/EGFR; A431 xenografted	Bind to EGFR, inhibit EGFR, AKT and ERK1/2 phosphorylation; in vivo significantly inhibit tumor growth; anti-proliferation	-	0–6.4 μM	[39]
		Resveratrol	Phytoalexin	*Diverse plants*	Prostate cancer DU145 (AR-)	Bind to EGFR, inhibit EGFR, AKT and ERK phosphorylation; anti-proliferation	-	10–50 μM	[40]
		Honokiol	Phenols	*Magnolia officinalis*	NSCLC; HNSCC xenograft	Inhibit EGFR, p-AKT and p-STAT3 expression; in vitro combination of erlotinib to inhibit invasion and metastasis; in vitro combination of honokiol and cetuximab to significantly enhance growth inhibition; (intraperitoneal injection)	100% ethanol	0.01–100 μM	[41]
		Genistein	Isoflavonoid	*Glycine max*	HSC3 and KB oral squamous cell carcinoma; OSCC xenograft	Downregulate p-EGFR and p-AKT in HSC3, but not in KB cells; in vivo combination with cetuximab to significantly enhance growth inhibition (IP injection); anti-proliferation	DMSO	0–80 μM	[42]
		Silibinin	Polyphenolic	*Silybium marianum*	NSCLC and 293T; PC-9 xenografts	Inhibit EGFR activity by interfering with dimerization of EGFR; inhibit EGFR expression and EGFRAKT phosphorylation; in vitro combination with silibinin and erlotinib to induce apoptosis and growth inhibition (oral gavage); anti-proliferation	-	0–200 μM	[43]
		Epigallocatechin-3-gallate	Flavonoid	*Camelia sinensis*	YCU-N861 and YCU-H891 Head and neck squamous cell carcinoma	Inhibit p-EGFR, p-STAT3 and p-ERK expression; G1 cell cycle arrest, induce apoptosis; combination with 5-fluorouracil to significantly enhance growth inhibition	Water	10–40 μg/mL	[44]
	EGFR/HER2	Berberine	Isoquinoline -alkaloid	*Huanglian and Huangbai*	Human ovarian cancer cells; HT-29 cell xenograft	Downregulate EGFR and HER2 expression; inhibit EGFR-HER2/PI3K/Akt signaling pathway; inhibit migration and invasion	DMSO	0–100 μM	[45]
	VEGFR2	Voacangine	Alkaloid	*Voacanga* *africana*	Glioblastoma HUVEC; U87MG cells xenograft	Bind to VEGFR2; inhibit p-VEGFR2, p-ERK and P-AKT expression; in vitro combination with sunitinib to inhibit growth; (IP injection); anti-proliferation;	DMSO	50 μM	[46]
		Ellagic acid	Polyphenol	*Pomegranate*	HUVECs and MDA-MB-231; Breast cancer xenograft	Bind to VEGFR2; inhibit p-VEGFR2, p-ERK, P-AKT and p-JNK expression; inhibit proliferation and invasion; in vivo inhibit growth (IP injection)	DMSO	0–20 μM	[47]
					Bladder cancer T24, UM-UC-3, 5637, HT-1376 and UM-UC-3 xenograft	Downregulate VEGFR2 expression; combination with mitomycin C to inhibit proliferation, migration and invasion; in vivo inhibit tumor growth; (IP injection)	DMSO	5–60 μM	[48]
		Matrine	Alkaloid	*Sophora flavescens*	UBC cell lines	Inhibit VEGFR2 BCL-2, caspase-3, p-AKT and p-PI3K expression; increase Bax and cleaved caspase-3; combination with cisplatin to inhibit proliferation, migration and invasion	Physiological saline	0–16 μM	[49]
	FLT3	Isoliquiritigenin	Flavonoid	*Licorice*	AML; AML xenograft model	Bind to FLT3; inhibit P-FLT3, p-STAT3, p-Erk1/2 and csapase3 expression; in vivo inhibit proliferation and induce apoptosis; (administered orally)	DMSO	0–40 μM	[50]
	EphA2R	Ellagitannin -metabolite urolithin D	Polyphenol	*Pomegranate*	Prostate cancer cells	Bind to EphA2R; inhibit EphA2 phosphorylation; anti-proliferation	DMSO	0–30 μM	[51]
	c-Met	Quercetin	Flavonoid	*Sophora japonica*	Melanoma	Reduce c-Met homo-dimerization; inhibit c-Met, Gab1, FAK and PAK phosphorylation; inhibit migration and invasion	DMSO	0–80 μM	[52]
G Protein-coupled receptors	PAR2	Epigallocatechin-3-gallate	Flavonoid	*Camelia sinensis*	Human colon cancer SW620	Inhibit that PAR2 activation by PAR2-AP or the TF/factor VIIa complex; inhibit ERK-1/2 and NF-κB activation; inhibit migration and proliferation;	-	0–100 µg/mL	[53]
	EP2	Silymarin	Polyphenolic	*Silybium marianum*	Human renal carcinoma Caki	Inhibit migration; inhibit EP2, p-CREB, p-STAT3, P-Src expression	-	20–50 µg/mL	[54]
	EP1	Epigallocatechin-3-gallate	Flavonoid found	*Camelia sinensis*	HCC cell line	Inhibit migration and proliferation; inhibit EP1 expression;	-	12.5–100 μg/mL	[55]
	EP2/EP4	Frondoside A	Triterpene glycoside	*Cucumaria frondosa*	Balb/cByJ female mice	Selectively antagonize EP2/4; inhibit p-Erk expression; (IP injection); inhibit metastasis	PBS	0.1–5 μM/L	[56]
	CB1/ CB2	Tetrahydrocannabinol and cannabidiol	Phenols	*Cannabis*	Glioblastoma cell lines	Bind to CB1 and/or CB2 receptors; induce CB1 and/or CB2 receptor activation; inhibit invasiveness; induce apoptosis	-	1.7 μM and 0.4 μM	[57]
	CB1	Quercetin	Flavonoid	*Sophora japonica*	Caco2 and DLD-1	Induce CB1-R expression; inhibit PI3K/AKT/mTOR and induce JNK/JUN pathways; inhibit the migration; induce apoptosis	DMSO or Absolute ethanol	10–50 μM	[58]
	Frizzled-7				BEL/5-FU and BEL-7402	Suppress FZD7 and β-catenin expression; decrease ABCB1, ABCC1 and ABCC2 expression; reverse MDR	-	0–160 μM	[59]
	Frizzled-8	Brucine	Alkaloid	*Strychnos nux-vomica L.*	LoVo, SW480 and Caco-2; xenograft model	Downregulate Frizzled-8, MMP2, MMP3 and MMP9; inhibit Wnt/β-catenin signaling pathway; inhibit migration, and induce apoptosis; (orally administered)	-	0–50 μM	[60]
Nuclear receptor superfamily	ERβ	Ellagic acid	Polyphenol	*Pomegranate*	HeLa and MCF-7	Estrogen antagonist	-	10^−6^–10^−8^ M	[61]
	AR	Luteolin	Flavonoid found	*Peanut*	Prostate cancer cell; LNCaP xenograft	Anti-proliferative; inhibit AR and PSA expression; in vivo inhibit tumor growth and downregulate AR and PSA (IP injection)	DMSO	0–40 μM	[62]
	ESRRG	Sophoridine	Monomeric alkaloid	*Sophora alopecuroides* L.	Gastric cancer AGS and SGC7901	Combination with cisplatin to inhibit proliferation, induce apoptosis; inhibit the migration and invasion; Enhance ESRRG expression and β-catenin degradation	DMSO	3 μM	[63]
Other receptors	DR5	Curcumin	Diketones	*Curcuma species*	Human renal Caki cells	Enhance ROS and DR5 expression; induce apoptosis;	-	0–30 μM	[64]
		Ginsenoside C-K	Saponins	*Ginseng*	Colon cancer cells	induce apoptosis, induce Bax, Bid, cytochrome c expression; inhibit MCL-1, BCL-2, cFLIP and XIAP expression; enhance ROS, DR5 and p-JNK expression	DMSO	0–100 μM	[65]
	PPARγ	Ginsenoside F2	Saponins	*Ginseng*	Mouse embryo fibroblast 3T3-L1	Bind to PPARγ; inhibit PPARγ and perilipin expression	-	10–100 μM	[66]
		Thymoquinone	-	*Nigella sativa*	Breast cancer cells	Bind to PPARγ; increase PPARγ activity; induce apoptosis	-	0–80 μM	[67]
		6-Shogaol	-	*Rhizomes of ginger*	Breast and colon cancer cells	Binding to PPARγ; increase in PPARγ activity; induce apoptosis	DMSO	0–100 μM	[68]
	TLR4	Curcumin	Diketones	*Curcuma species*	RAW264.7	Inhibit dimerization of TLR4	-	10–50 μm	[69]

## Data Availability

Not applicable.

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
