# Peer review of "Small Molecule Compounds of Natural Origin Target Cellular Receptors to Inhibit Cancer Development and Progression"

_ijms, 2022, doi:10.3390/ijms23052672_

Round 1
Reviewer 1 Report
In the proposed review, Wang and colleagues summarized the available information about the small natural compounds used to combat cancer diseases influencing cell signaling via different receptors. They tried to group the target receptors according to the nature of the transported substances respectively natural products by providing information about the components’ classification, the sources from which they were isolated, pharmacological activity, doses used and reference. The mechanisms of influencing of the target receptors by the small natural compounds are nicely illustrated with a figure. The conclusions are logical and clear.
The review will be a starting point for many future studies related to the antitumor therapy.
Reviewer 2 Report
In this review article, the authors revise some natural-derived compounds available for targeting different types of receptors susceptible to be dysregulated in cancer. The approach they take, separating among receptor types, is very interesting. Moreover, they give an overview of diverse natural compounds that could catch the attention of the scientific community and be useful for cancer treatment in the future, when proper studies have been performed. I also like how the review is summarized in the Figure 1 and Table 1.
Nonetheless, despite it is interesting to know the existence of all those compounds for future studies, I think the review might be a bit superficial. The authors mention a lot of different molecules and some of the results of studies that proved its mechanisms of action, but in my opinion they do not cover appropriately aspects such as results in different models (in vitro and in vivo), synergies with other established or novel anti-cancer compounds, previous usage with other purposes (i.e. palliative purposes for secondary effects of chemotherapy) and a discussion about the future implementation of these molecules in therapies based on the mentioned studies.
For the review to be more useful, the authors should focus on the most important compound(s) in each category of receptors (i.e. resveratrol) and extensively develop the aspects I mentioned and discuss a bit its potential for future therapies. Then, for the rest of the compounds, they can give an overview as they made.
Other question that I think it is not clear is the criteria the authors followed for the selection of the compounds, because there are a lot of more natural compounds that affect the discussed receptors that are not included. For example, they mention the cannabinoid receptors (paragraph from 368-376), but only mention the “quercetin” as modulator and not phytocannabinoids as CBD or THC, which have proven to have anti-tumoral properties in different types of cancer.
In general, I like the idea of this review but for being really useful the authors should make substantial changes in the approach and the content, trying to not only superficially mention natural compounds and some related studies but really justify and discuss the usefulness in the future treatment of oncologic patients.
Round 2
Reviewer 2 Report
The authors have addressed almost all the issues I pointed in the first revision. The most important is that they have included the criteria followed for selecting the described compounds and they have extended some descriptions that I considered important for the revision to be complete (i.e. extended description of the most important natural compounds and the synergies).
In this review, the authors have given more importance to cover a larger number of compounds than to extensively describe the use of each of them. Nonetheless, with the introduced corrections they cover the important aspects of the ones that could be more useful to date.
Finally, I like the paragrpahs introduced in each section to highlight the potential use of these compounds in clinics.
Overall, I think that this review will be useful for the scientific community to know the potential of natural compounds in cancer treatment.